**Data Availability Statement:** All relevant data are within the paper and its Supporting Information files.

**Funding:** This study was supported by the National Special Science and Technology Project for Major

# Spatial-temporal analysis of pulmonary tuberculosis in Hubei Province, China, 2011–2021

**Yu Zhang** 📷, **Jianjun Ye, Shuangyi Hou, Xingxing Lu, Chengfeng Yang, Qi Pi, Mengxian Zhang, Xun Liu, Qin Da, Liping Zhou** 📷 *

Department of Tuberculosis Control and Prevention, Hubei Provincial Center for Disease Control and Prevention, Wuhan, Hubei, China

* zhouliping2268@163.com

## Abstract

### Background

Pulmonary tuberculosis (PTB) is an infectious disease of major public health problem, China is one of the PTB high burden counties in the word. Hubei is one of the provinces having the highest notification rate of tuberculosis in China. This study analyzed the temporal and spatial distribution characteristics of PTB in Hubei province for targeted intervention on TB epidemics.

### Methods

The data on PTB cases were extracted from the National Tuberculosis Information Management System correspond to population in 103 counties of Hubei Province from 2011 to 2021. The effect of PTB control was measured by variation trend of bacteriologically confirmed PTB notification rate and total PTB notification rate. Time series, spatial autonomic correlation and spatial-temporal scanning methods were used to identify the temporal trends and spatial patterns at county level of Hubei.

### Results

A total of 436,955 cases were included in this study. The total PTB notification rate decreased significantly from 81.66 per 100,000 population in 2011 to 52.25 per 100,000 population in 2021. The peak of PTB notification occurred in late spring and early summer annually. This disease was spatially clustering with Global Moran's $I$ values ranged from 0.34 to 0.63 ($P<0.01$). Local spatial autocorrelation analysis indicated that the hot spots are mainly distributed in the southwest and southeast of Hubei Province. Using the SaTScan 10.0.2 software, results from the staged spatial-temporal analysis identified sixteen clusters.

### Conclusions

This study identified seasonal patterns and spatial-temporal clusters of PTB cases in Hubei province. High-risk areas in southwestern Hubei still exist, and need to focus on and take targeted control and prevention measures.

Infectious Diseases of China (Grant No. 2017ZX10302301-005-002). The funders had no role in study design, data collection and analysis, decision to publish, or preparation of the manuscript.

**Competing interests:** The authors have declared that no competing interests exist.

# Background

Tuberculosis (TB) is a chronic infectious disease mainly transmitted through the respiratory tract. It is the main cause of death from single pathogen and a major public health and social problem in the world [1]. The ambitious goal for global TB control is to achieve the United Nations Sustainable Development Goals by 2030 [2] and the Strategic goal of ending the TB epidemic by 2035 [3]. There is still a huge gap between the current status of global TB control and the goal of ending the TB epidemic. China is the second country with the highest TB burden, with an estimated 842,000 new TB cases in 2020, accounting for 8.5% of the global total, and 30,000 TB deaths [1].

Hubei province is located in the middle of China and is one of the provinces with high prevalence of pulmonary tuberculosis (PTB) in China [4]. The annual case number of PTB ranks the second high in class A and B infectious diseases in Hubei Province. The reported incidence of PTB in Hubei province had always been over 75 cases per 100,000 population before 2016 [5], which was higher than the national average. There were regional differences in the distribution of tuberculosis, the highest notification rate of Badong county (152.77 per 100,000 population) was 6.11 times higher than the lowest rate of Dongbao district (25.01 per 100,000 population) in 2021.

In recent years, spatial epidemiology, as a branch of epidemiology, has been widely used in disease control and prevention [6, 7]. Studies in China [8–10] and other countries [11–13] confirmed that TB has highly complex dynamics and spatially heterogeneous at the national level and the provincial level [14–18]. Few studies have been conducted in Hubei province to explore the spatial epidemiology at the county level. In order to improve tuberculosis control measures and set priorities and targets effectively, we conducted a spatial-temporal cluster analysis of the notification of PTB at county and district levels in Hubei Province from 2011 to 2021.

# Methods

## Study setting

Hubei province is located in the middle reaches of the Changjiang River and north of Dongting Lake, with 13 cities and 103 districts/counties. It covers an area of 185,900 square kilometers, of which 56% are mountains, 24% hills and 20% plain lakes [19]. In 2021, the permanent resident population was 57.75 million, and the urban population was 36.32 million, accounting for 62.89%. The total GDP was RMB 5001.294 billion yuan in 2021 [20].

## Data source

The active PTB numbers registered from 2011 to 2021 in Hubei province were extracted from the National Tuberculosis Information Management System (TBIMS) [21]. The TBIMS serves as the national TB surveillance system, which is established and operated by the Chinese Center for Disease Control and Prevention (CCDC). It contains demographic and clinical information such as age, sex, occupation, address, date of diagnosis, results of smear microscopy, type of TB, history of TB treatment, and treatment outcomes. Sensitive information such as names of patients and phone numbers were excluded in this study because of privacy and confidential issues. We obtained the TB data from Hubei Tuberculosis Control Institute, and were permitted to use by CCDC.

The definition of active PTB was according to the Health Standard of the People's Republic of China WS196–2017 [22]. This study included all forms of PTB, including bacteriologically confirmed and clinically diagnosed PTB, previously treated and new PTB, and childhood and adult PTB.

The annual population data of each administrative district from 2011 to 2018 were obtained from the Hubei Statistical Yearbook and the Basic Information System for Disease Control and Prevention.

## Ethical statement

This study was approved by the Ethics Committee of Hubei Center for Disease Control and Prevention. There was no access to individual information, including name, identity information, address, telephone number, etc. This study was a secondary analysis of the data on county-level, therefore, waiver for informed consent was sought.

## Data analysis

**Descriptive analysis.** This study was based on the active PTB notification rates in the whole province. SPSS version 25.0 (SPSS, IBM; Inc., Chicago, IL, USA) was used for analysis and the statistically significant level was $\alpha$ = 0.05. ArcGIS v.10.3 software (ESRI Inc., Redlands, CA, USA) was used to create a spatial database and carry out spatial autocorrelation analysis.

**Spatial autocorrelation analysis.** Spatial autocorrelation refers to the correlation of the same variable in different spatial locations, which is a measure of the aggregation degree of spatial unit attribute values. The basic measure of spatial autocorrelation analysis is spatial autocorrelation coefficient, which is usually divided into global spatial autocorrelation analysis and local spatial autocorrelation analysis.

Global Moran's $I$ [23, 24], a global test statistics for spatial autocorrelation, is used to identify spatial autocorrelation of tuberculosis in Hubei province. The value of Global Moran's $I$ varies between -1 and 1. A positive Moran's $I$ value indicates a positive correlation, and the larger the value, the stronger the correlation. In contrast, a negative Moran's $I$ value indicates a negative correlation, showing a discrete distribution. When the value is 0, there is no spatial clustering, meaning that the data are randomly distributed [25, 26]. Its statistical significance was tested according to the standardized statistic $Z$-score.

Local Getis's $Gi^*$ statistic [27] is a local spatial autocorrelation indicator, which can measure the aggregation degree of high or low values, detect hot spots and cold spots where events occur, and test its statistical significance with the standardized statistic Z ($G_i$). Value of $Gi^*$ in this study was used to test its statistical significance.

**Spatial-temporal scan analysis.** The space-time scan statistics proposed by Kulldorff include time, space and space-time scan statistics and have been widely used in many fields [28]. The SaTScan™ version 10.0.2 software was developed by Martin Kulldorff together with Information Management Services Inc [29]. and could download on Web (https://www. satscan.org/), which is used for spatial-temporal scanning analysis. The method is based on a moving column scanning window, with the base of the column corresponding to geographic region and the height of the column corresponding to time. For each scanning window, the theoretical incidence can be calculated according to the total incidence and the number of people in the window, and then the logarithmic likelihood ratio (LLR) is constructed by using the actual incidence and theoretical incidence inside and outside the scanning window respectively. The window with the largest LLR is selected from all the scanning Windows, which is the window with the strongest clustering. The secondary clusters are the other windows with statistically significant LLR value. In order to more accurately describe the characteristics of spatial clustering distribution of TB notification rate, based on spatial autocorrelation, this study divided the spatial-temporal analysis time into two stages 2011–2015 (the Twelfth Five-year Plan) and 2016–2021 the Thirteenth Five-Year Plan, as the Fourteenth Five-Year Plan has not been released, the analysis time was not listed separately in 2021), according to the

planning period of TB prevention and control in Hubei Province. The maximum spatial scanning area was set as 40% of the total population of the province. Monte Carlo simulation test was used to evaluate whether the difference is statistically significant, ArcMap software was used to visualize the scanning results.

## Results

### Descriptive analysis of PTB cases

Table 1 showed the demographic characteristics and clinical characteristics of PTB cases. There was a total of 436,955 PTB cases were registered in Hubei from 2011 to 2021, of which 310,071 (70.96%) were male and 126,884 (29.04%) were farmers by occupation. A significant proportion of TB cases happened among adults between the age of 25 to 44 years (about 25%)

**Table 1. Demographic and clinical characteristics of PTB patients in Hubei Province, 2011–2021.**

| Variables | Number | Percent |
|---|---|---|
| Sex | | |
| Male | 310,071 | 70.96 |
| Female | 126,884 | 29.04 |
| Age group, y | | |
| <25 | 58,095 | 13.30 |
| 25–44 | 109,952 | 25.16 |
| 45–64 | 170,114 | 38.93 |
| ≥65 | 98,794 | 22.61 |
| Occupation | | |
| Farmer | 269,842 | 61.76 |
| Student | 19,891 | 4.55 |
| Laborer | 22,419 | 5.13 |
| Retired | 20,945 | 4.79 |
| Government employee | 6,490 | 1.49 |
| Non-government employee | 12,838 | 2.94 |
| Unemployed | 56,862 | 13.01 |
| Medical staff | 2,044 | 0.47 |
| Others | 25,624 | 5.86 |
| Patient sources | | |
| Contact check | 590 | 0.14 |
| Recommended due to symptoms | 8,498 | 1.94 |
| Seek medical treatment | 151,440 | 34.66 |
| Health examination | 7,001 | 1.60 |
| Referral | 185,497 | 42.45 |
| Track | 82,815 | 18.95 |
| Other | 1,114 | 0.26 |
| Treatment history | | |
| New case | 409,699 | 93.76 |
| Retreated case | 27,256 | 6.24 |
| Type of tuberculosis | | |
| Bacteriologically confirmed | 190924 | 43.69 |
| Bacteriological negative | 239778 | 54.87 |
| No bacteriological results | 433 | 0.10 |
| Tuberculosis pleurisy | 5820 | 1.33 |

and 45 to 64 years (over 38%). Approximately one third (34.66%, 151,440) of the patients were diagnosed when symptomatic individuals come to seek treatment at health care facilities, and 185,497 (42.45%) cases were referral patients. Overall, 409,699 (93.76%) of the cases were new cases, and 190,924 (43.69%) cases were bacteriologically confirmed.

## Temporal patterns of PTB cases

The annual average notification rate of all PTB and bacteriologically confirmed PTB were 67.97 per 100,000 population and 29.70 per 100,000 population, respectively. The total PTB notification rate decreased significantly from 81.66 per 100,000 population in 2011 to 52.25 per 100,000 population in 2021 ($\chi 2$ trend = 10101.80, $P < 0.001$) (Fig 1). The monthly PTB notification rates showed a trend of volatility and decline from 2011 to 2021 in Hubei Province. PTB cases have shown seasonal variations with the highest number of PTB cases registered in late spring and summer (March–June) in all years. Due to the impact of COVID-19, the number of registered TB patients dropped sharply in February 2020, with fewer than 1,000 cases in the province, and the notification was recovered by April 2020 (Fig 2).

## Spatial patterns of PTB cases

The areas with high PTB notification rates are southwest and southeast Hubei province, mainly located in Badong county, Xuan 'en County, Xianfeng County, Jianshi County of Enshi Prefecture and Tongshan County and Jiayu County of Xianning CityThe results of global spatial autocorrelation analysis of PTB notification rate in Hubei province from 2011 to 2021 showed that Moran's *I* value of each year was positive and the *P* values were less than 0.001, and the autocorrelation was statistically significant in regional scope, suggesting that there was a spatial positive correlation between PTB notification rate in Hubei (Table 2).

Local G statistics were used for local autocorrelation test, and the results showed that hot spots and cold spots existed in PTB notification rate in Hubei Province from 2011 to 2021. The

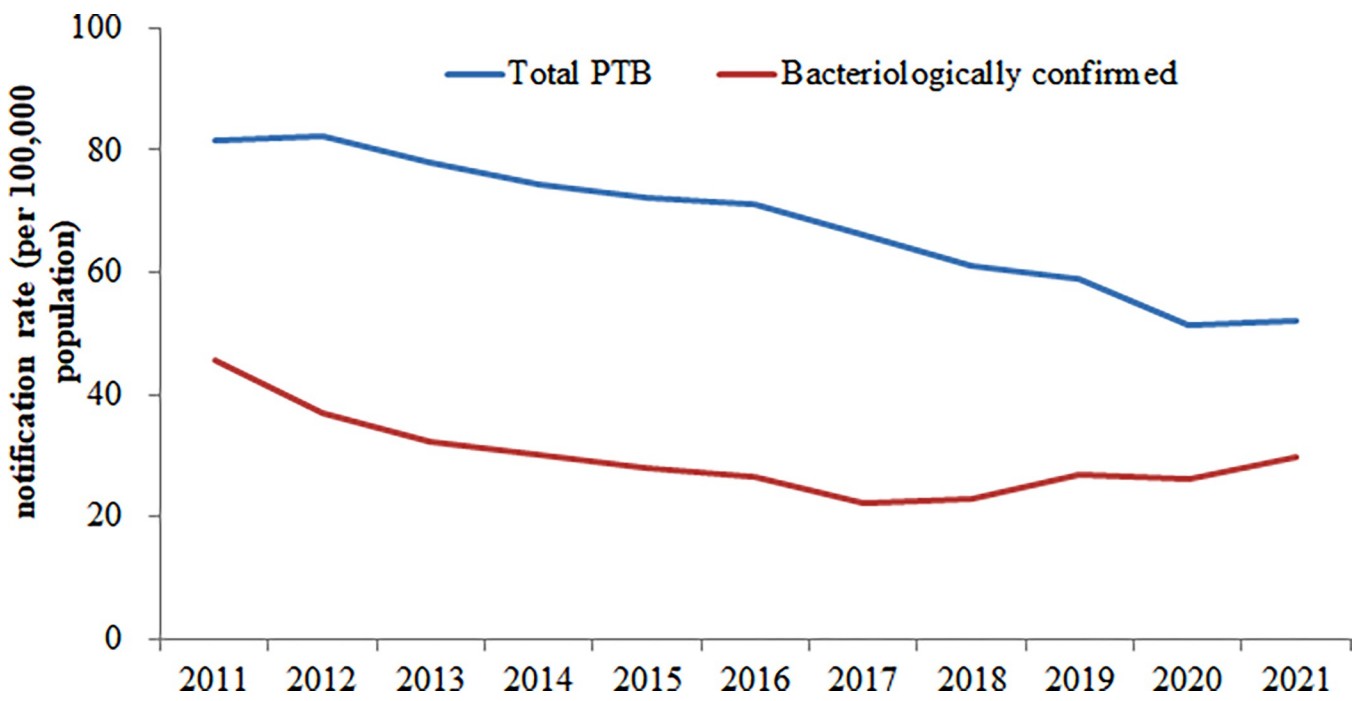

**Fig 1. The variation trend of TB notification rate from 2011 to 2021 in Hubei.**

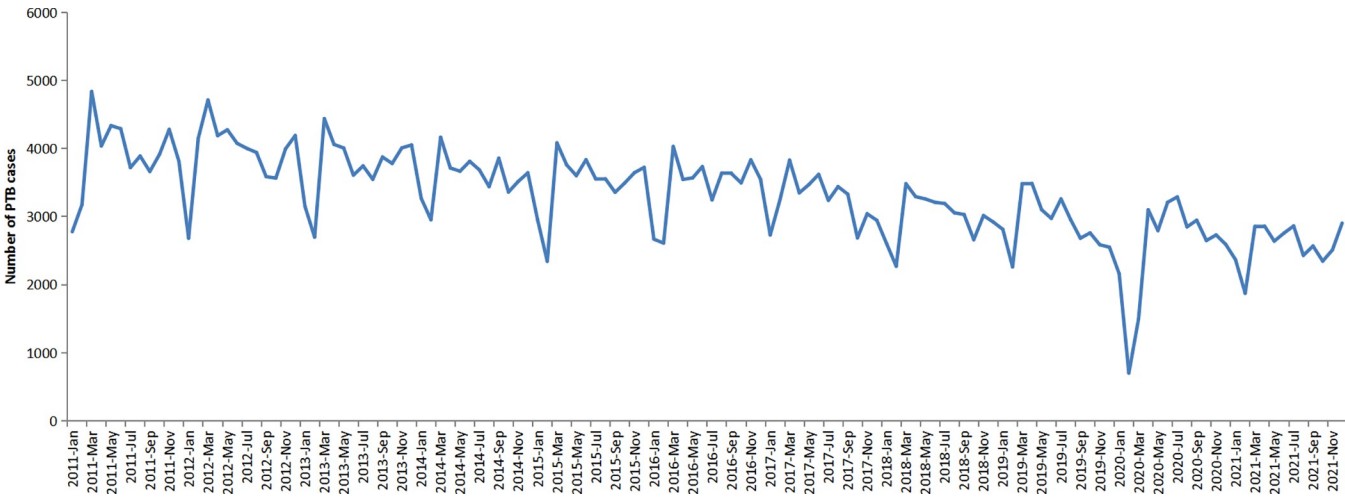

**Fig 2. The monthly registered number of PTB cases in Hubei Province, China, 2011–2021.**

hot spots are mainly distributed in the southwest and southeast of Hubei Province, including Enshi prefecture and some counties and districts of Yichang city, and Tongshan county, Xian'an district, and Jiayu county of Xianning city. This indicates that tuberculosis notification rates are higher in these areas. Some hot spots have changed dynamically over time. From 2011 to 2021, the number of counties covered by the hot spot area southwest Hubei increased year by year, only slightly decreased in 2016, and all counties in Enshi prefecture, Shennongjia region and three counties in Yichang city were covered in 2021. However, the hot spots in southeast Hubei gradually decreased and disappeared in 2021. On the other hand, the cold spots are mainly distributed in central and northern Hubei, including some counties of Jingmen city, some counties of Xiangyang city and Suizhou city.

## Spatial-temporal clustering analysis by SaTScan

Spatial-temporal clustering analysis by SaTScan indicated that the notification rates of PTB were spatial-temporal clustered. The results showed in Table 3. In the first stage, ten clusters appeared from 2011 to 2015, among which the most likely clusters covered eight counties and

**Table 2. Global spatial autocorrelation analysis of PTB notification rate in Hubei Province, China from 2011 to 2021.**

| Year | Moran's *I* | *Z*-score | *P* value |
|------|-------------|-----------|-----------|
| 2011 | 0.447 | 7.234 | <0.001 |
| 2012 | 0.478 | 7.737 | <0.001 |
| 2013 | 0.421 | 6.807 | <0.001 |
| 2014 | 0.405 | 6.564 | <0.001 |
| 2015 | 0.425 | 6.909 | <0.001 |
| 2016 | 0.335 | 5.474 | <0.001 |
| 2017 | 0.400 | 6.533 | <0.001 |
| 2018 | 0.524 | 8.483 | <0.001 |
| 2019 | 0.525 | 8.546 | <0.001 |
| 2020 | 0.627 | 10.121 | <0.001 |
| 2021 | 0.570 | 9.315 | <0.001 |

**Table 3. Space-time clusters of PTB in Hubei province, China detected by SaTScan from 2011 to 2021.**

| Time period | Cluster Type | Number of Clustering areas | Cluster districts and counties | Time frame | Observed cases | Expected cases | Relative risk | Log likelihood ratio | P value |
|---|---|---|---|---|---|---|---|---|---|
| 2011–2015 | Most likely cluster | 8 | Lichuan, Xianfeng, Enshi,Xuanen, Laifeng, Jianshi, Hefeng, Badong | 2011–2012 | 8027 | 5130.40 | 1.59 | 715.76 | <0.001 |
| | Secondary cluster 1 | 6 | Tongcheng, Chongyang, Chibi, Jiayu, Tongshan, Xian'an | 2012–2013 | 5623 | 3831.57 | 1.48 | 372.81 | <0.001 |
| | Secondary cluster 2 | 6 | Huangmei, Wuxue, Qichun, Xisaishan, Yangxin, Xishui | 2011–2012 | 8186 | 6502.13 | 1.27 | 207.87 | <0.001 |
| | Secondary cluster 3 | 7 | Macheng, Hongan, Luotian, Tuanfeng, Xinzhou, Huangpi, Yingshan | 2011–2012 | 8043 | 6908.58 | 1.17 | 91.39 | <0.001 |
| | Secondary cluster 4 | 1 | Yuan'an | 2013–2014 | 506 | 288.75 | 1.75 | 66.71 | <0.001 |
| | Secondary cluster 5 | 2 | Yunmeng, Yingcheng | 2011–2012 | 2212 | 1740.23 | 1.27 | 59.35 | <0.001 |
| | Secondary cluster 6 | 1 | Danjiangkou | 2011–2012 | 897 | 691.33 | 1.30 | 28.04 | <0.001 |
| | Secondary cluster 7 | 1 | Jiangxia | 2011–2012 | 1359 | 1113.84 | 1.22 | 25.33 | <0.001 |
| | Secondary cluster 8 | 1 | Caidian | 2013–2014 | 843 | 674.21 | 1.25 | 19.62 | <0.001 |
| | Secondary cluster 9 | 5 | Duodao, Shayang, Dongbao, Dangyang, Zhongxiang | 2012–2013 | 4620 | 4215.61 | 1.10 | 19.18 | <0.001 |
| 2016–2021 | Most likely cluster | 14 | Hefeng, Wufeng, Xuanen, Jianshi, Enshi, Changyang, Badong, Laifeng, Xianfeng, Zigui, Yidu, Dianjun, Lichuan, Xiaoting | 2017–2019 | 14287 | 8799.54 | 1.67 | 1511.19 | <0.001 |
| | Secondary cluster 1 | 6 | Tongcheng, Chongyang, Chibi, Jiayu, Tongshan, Xian'an | 2016–2018 | 7190 | 4569.47 | 1.59 | 655.21 | <0.001 |
| | Secondary cluster 2 | 20 | Yingshan, Luotian, Xishui, Qichun, Tuanfeng, Macheng, Huangzhou, Xisaishan, Wuxue, Xinzhou, Huangshigang, Huangmei, Echeng, Xialu, Huarong, Tieshan, Hongan, Daye, Yangxin, Liangzihu | 2016–2017 | 15142 | 12964.85 | 1.18 | 185.25 | <0.001 |
| | Secondary cluster 3 | 1 | Caidian | 2016–2018 | 1212 | 847.96 | 1.43 | 69.19 | <0.001 |
| | Secondary cluster 4 | 6 | Zhongxiang, Dongbao, Yicheng, Duodao, Jingshan, Shayang | 2016 | 2597 | 2067.68 | 1.26 | 63.28 | <0.001 |
| | Secondary cluster 5 | 1 | Xiaonan | 2016–2017 | 1480 | 1116.88 | 1.33 | 53.82 | <0.001 |

districts, mainly distributed in Enshi Prefecture, southwest Hubei province, and clustered during 2011 to 2012. The nine secondary clusters covered 30 counties and districts, mainly distributed in southeast hubei and eastern Hubei, including Xianning City, Huanggang city and part of the far urban area of Wuhan city, as well as the central plain area. The mainly clustering time was from 2012 to 2013. In addition, Yuan'an county in western Hubei and Danjiangkou county-level city in northern Hubei were clustered in a single county. In the second stage, six clusters appeared from 2016 to 2021. The most likely clusters covered fourteen counties, which added six counties in Yichang city adjacent to Enshi Prefecture on the basis of the first stage. The clustering time was from 2017 to 2019 with 50,596 cases were registered. The analysis results showed that the risk of PTB in these districts and counties was 1.67 times higher than that outside the hot spots. The five secondary clusters covered 34 counties, which added some counties and districts in Huangshi city in eastern Hubei, and others were roughly the same as the first stage distribution. The mainly clustering time was from 2016 to 2017. Danjiangkou

city in northern Hubei, which had a single county cluster in the first stage, did not appear in the second stage.

## Discussion

In our study, descriptive analysis showed that the notification rate of PTB in Hubei province declined steadily during the eleven-year study period. The PTB notification rate dropped from 81.66 cases per 100,000 population in 2011 to 52.25 cases per 100,000 population in 2021. This downward trend is consistent with other provincial and national studies [8, 15–18]. The decline is related to the great attention paid by the Hubei provincial government and health administrative department to TB control and prevention in recent years. The Hubei Provincial Tuberculosis Control and Prevention Plan (2011–2015) was issued in May 2012 by the Hubei Provincial government with a series of measures, which mainly requiring greater efforts to detect patients. Subsequently, during the Thirteenth Five-Year Plan (2016–2020) for TB and control and prevention, an integrated system under the collaboration of Disease Control and Prevention, tuberculosis-designated hospitals, and primary health centers, which was established with a clear division of labor and coordination among the institutions. In 2019, the Hubei Provincial Health Administrative Department has purchased a batch of molecular biological testing equipment for TB diagnosis and treatment and distributed them to designated TB medical institutions at all levels at county and district levels in the province. The application of new diagnostic technology shortened the diagnosis time of PTB and improved the percentage of cases confirmed bacteriologically. At the same time, the screening of rifampicin-resistant PTB was further strengthened. With the implementation of these effective measures, the TB epidemic situation in Hubei province declined steadily. Although significant efforts made an annual decline rate of 4.37% from 2011 in Hubei province, the high TB burden with absolute number of 30,970 cases registered in 2021, made a huge challenge to achieve the goal of End TB in 2035 without the breakthrough of vaccine or new drug [3, 30].

Through the time series study, seasonal trends were detected with apparent peaks in late spring and summer. In Hubei Province, the winter is from December to February, and the coldest months is January. Reduced exposure to ultraviolet rays from sunlight and poor ventilation in the indoor environment in winter may increase the chance of transmission among the infectious source and the close contacors [31–36]. In addition, People were busy celebrating the Lunar New Year during the Spring Festival and avoiding to seek medical care, leading to a significant decrease in TB reports during the holiday, which mainly in February. The impact of delayed diagnosis should also be considered. In rural and remote areas of Hubei Province, most patients chose the nearest primary health facilities for their first medical consulation rather than visited TB designated hospitals directly, but the primary health facilities usually have no capacity or poor diagnosis of TB, and may misdiagnosed patients with common diseases such as cold or fever, leading to delayed in TB diagnosis [36], and consequently, more PTB cases were reported in summer seasons. Such seasonal patterns identified were consistent with previous studies in many regions such as in China [35], Wuhan city [37], Xingjiang autonomous prefecture of China [38], Yunnan province [30], Indian [39].

The global spatial autocorrelation results of this study indicated that the PTB notification rate in Hubei province presents an obvious spatial clustering distribution, which is consistent with the results of previous short-term spatial analysis [40]. Advanced local spatial autocorrelation analysis showed that the hot spots of PTB had a slight dynamic change with time. In this study, the TB hotspots were basically consistent with the areas with high TB notification rates in Hubei Province. Enshi Prefecture in southwest Hubei province (Badong county, Xianfeng County, Hefeng County, Xuan 'en County, Laifeng County, Jianshi County), Yichang city

(Changyang County, Yuan 'an County, Xingshan County) and Tongshan County in Xianning city in southeast Hubei province were the areas with high notification rate of PTB in the past 11 years. These regions ranked among the top 10 in average annual notification rates from 2011 to 2021, and were also at risk of TB transmission. The following facts may explain why these counties became hot spots. First of all, Enshi Tujia and Miao Autonomous Prefecture in southwest Hubei is a multi-ethnic residence, adjacent to Xiangxi Tujia Autonomous Prefecture of Hunan Province, Qianjiang District and Wanzhou District of Chongqing. Enshi prefecture is mostly mountainous, subtropical monsoon mountainous humid climate, its GDP ranked behind in Hubei province. The characteristics of the relatively backward economic conditions and inconvenient transportation restricted the investment of tuberculosis control funds and professionals, and further affected the implementation of local tuberculosis control plans. In addition, TB patients have discontinued treatment due to financial difficulties, or infection control measures were not in place, leading to the continued spread of the disease. These situations also occured in the neighboring counties of Changyang Tujia Autonomous County, Zigui County and Xingshan County, which were not all hot spots from 2011 to 2014. On the other hand, Xianning city, located in the southeast of Hubei province, is known as the southern gate of Hubei province, adjacent to Hunan province and Jiangxi Province, the local tourism resources are relatively rich. The study results detected that there were four counties and districts (Xian 'an District, Chibi City, Chongyang County and Tongshan County) as hot spots in Xianning City from 2011 to 2016, and the hot spots gradually decreased from 2017 to 2021, and disappeared in 2021. This indicates that the prevalence of tuberculosis in Xianning city has declined in recent years. This is due to the increased attention paid by local governments to TB control and prevention in recent years, which has been accompanied by a series of interventions along with increased funding. For example, Xianning city has carried out patient care work, improved the treatment compliance of patients. In addition, extensive and in-depth health promotion has been carried out to improve the awareness rate of TB among the whole population. However, although the epidemic situation in Xianning city has decreased, it is still a high prevalence area in Hubei Province, which still needs to be focused on.

Based on global and local spatial analysis, southwest and southeast Hubei were the key areas of tuberculosis prevention and control in Hubei Province. Considering the role of time in the geographical distribution of diseases, we used spatial-temporal scanning analysis to supplement the simple spatial analysis. In the previous analysis of spatial-temporal clustering of TB in China, we found that Hubei province was in the most likely cluster, and the main clustering time was 2005, 2007–2008, and 2011–2013, respectively [8, 15, 16]. This indicates that although the PTB epidemic in Hubei province is decreasing gradually, it is still at a high level nationwide. In this study, a staged spatial-temporal scanning analysis was applied, and a total of 16 clusters were found. Compared with the period of the Twelfth Five-year Plan, the PTB notification rate in Hubei province decreased by four clusters during in the period of Thirteenth Five-Year Plan. This indicated that Hubei Province has achieved certain results in the control and prevention of TB in recent years. However, from the perspective of cluster distribution, the most likely clusters in the second stage added six adjacent counties on the basis of the first stage. Obviously, the burden of PTB in southwestern Hubei was heavy and the risk of disease transmission was high. So these areas will be the most important areas for TB control and prevention in Hubei province in the next few years. In addition, the notification rate of some counties and districts in Yichang City, southwestern Hubei showed an upward trend from 2017 to 2019, which may be related to the investigation of PTB under-registration conducted in our province in 2015 [41]. The quality of PTB notification in these areas was improved, so that clusters can be detected. The secondary clusters were mainly located in eastern and southeastern Hubei, with relatively large populations. In previous studies, it was found

that the reported incidence of smear-positive PTB in these counties was clustered, indicating that there might be mutual transmission of PTB between these adjacent counties [40]. There may be potential for drug resistant clusters to spread. However, we did not perform this analysis because we did not have comprehensive drug susceptibility test data for all active TB patients. These identified clusters differed in their socio-economic status, demography structures, and natural environment. Effective and targeted measures should be taken to control TB transmission in these areas. The clusters identified by the spatial-temporal clustering analysis and the local spatial analysis were similar, which may indicate the robust of our analysis results.

However, this study was subject to some limitations. First of all, in order to improve the quality of surveillance data, we routinely conducted a sample survey of PTB notification every year, and conducted a large-scale survey every five years. However, it is still inevitable that a small number of PTB cases had not been captured, which might cause under-estimation of PTB epidemic in Hubei Province. Second, the present study only analyzed the spatial and temporal patterns of PTB cases and clusters. The high prevalence of PTB may be related to individual habits [42], socio-economic factors, medical conditions [43], and meteorological environment [44]. Further research is needed to reveal the role of these factors in the transmission of tuberculosis.

## Conclusions

This study identified temporal trends and spatial distribution of PTB cases at the county level in Hubei province from 2011 to 2021. PTB showed an annual downward trend in Hubei. The clustered areas of the staged spatial-temporal scanning had reduced, and progress had been made in TB control programs. High-risk areas in southwestern Hubei still exist, and need to focus on and take targeted control and prevention measures.

## Supporting information

**S1 Table.**
(DOCX)

## Author Contributions

**Conceptualization:** Jianjun Ye, Shuangyi Hou, Liping Zhou.

**Data curation:** Yu Zhang, Xingxing Lu, Qi Pi.

**Formal analysis:** Yu Zhang, Liping Zhou.

**Investigation:** Yu Zhang, Xingxing Lu, Chengfeng Yang, Mengxian Zhang, Xun Liu, Qin Da.

**Methodology:** Yu Zhang.

**Resources:** Liping Zhou.

**Writing – original draft:** Yu Zhang.

**Writing – review & editing:** Shuangyi Hou, Liping Zhou.

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
