## [Decision Letter · Decision Letter 0]

21 Oct 2022

PONE-D-22-10280

Spatial-temporal analysis of pulmonary tuberculosis in Hubei Province, China, 2011-2021

PLOS ONE

Dear Dr. Zhou,

Thank you for submitting your manuscript to PLOS ONE. After careful consideration, we feel that it has merit but does not fully meet PLOS ONE’s publication criteria as it currently stands. Therefore, we invite you to submit a revised version (Minor Revision) of the manuscript that addresses the points raised by the two reviewers.

We look forward to receiving your revised manuscript.

Kind regards,

Zhenlong Li, Ph.D.

Academic Editor

PLOS ONE

2.In ethics statement in the manuscript and in the online submission form, please provide additional information about the patient records/samples used in your retrospective study. Specifically, please ensure that you have discussed whether all data/samples were fully anonymized before you accessed them and/or whether the IRB or ethics committee waived the requirement for informed consent. If patients provided informed written consent to have data/samples from their medical records used in research, please include this information.

3.In your Data Availability statement, you have not specified where the minimal data set underlying the results described in your manuscript can be found. PLOS defines a study's minimal data set as the underlying data used to reach the conclusions drawn in the manuscript and any additional data required to replicate the reported study findings in their entirety. All PLOS journals require that the minimal data set be made fully available. For more information about our data policy, please see http://journals.plos.org/plosone/s/data-availability.

4.We note that [Figures 3-5] in your submission contain [map/satellite] images which may be copyrighted. All PLOS content is published under the Creative Commons Attribution License (CC BY 4.0), which means that the manuscript, images, and Supporting Information files will be freely available online, and any third party is permitted to access, download, copy, distribute, and use these materials in any way, even commercially, with proper attribution. For these reasons, we cannot publish previously copyrighted maps or satellite images created using proprietary data, such as Google software (Google Maps, Street View, and Earth). For more information, see our copyright guidelines: http://journals.plos.org/plosone/s/licenses-and-copyright.

a. You may seek permission from the original copyright holder of Figures 3-5 to publish the content specifically under the CC BY 4.0 license. 

Natural Earth (public domain): http://www.naturalearthdata.com/.

Reviewers' comments:

Reviewer's Responses to Questions

**Comments to the Author**

1. Is the manuscript technically sound, and do the data support the conclusions?

Reviewer #1: Yes

Reviewer #2: Yes

2. Has the statistical analysis been performed appropriately and rigorously? 

Reviewer #1: Yes

Reviewer #2: I Don't Know

3. Have the authors made all data underlying the findings in their manuscript fully available?

Reviewer #1: Yes

Reviewer #2: Yes

4. Is the manuscript presented in an intelligible fashion and written in standard English?

Reviewer #1: Yes

Reviewer #2: Yes

5. Review Comments to the Author

Reviewer #1: This manuscript clearly explained the TB endemic change of Hubei province using spatial-temporal analysis during ten years. However, there are some points need to provide more details.

1. In discussion part, the author thought the prevalence of TB in Hubei has been controlled to a certain extent. However, it could be biased if it was judged only by decline of notification rate. It maybe due to low detection of PTB cases. More data was needed to this conclusion.

2. For season trend, the author explained that most TB infections may occur in winter, with symptoms appearing after a certain time. However, for most adults in China, TB infection more likely occurred in previous years rather than recent one year. Thus this point was not reasonable. And also for “the impact of delayed diagnosis should also be considered”, what does it mean? Need much more explanation.

3. The study showed that Enshi Prefecture and Yichang County and Tongshan County in Xianning city were the areas with high notification rate of PTB in the past 11 years. Is there any change of hotspots for these regions during the 11 years? If there was some changes, it could be very useful to understand local TB control effects.

Reviewer #2: The authors report the epidemiology of tuberculosis in their region in relation to control measures and provide evidence that these efforts have been at least partially effective. The data presented is impressive and I have only minor comments.

In the limitation section line 341 – 351 I feel the authors should include the fact that as far as I am aware they do not have access to any genotyping data or drug resistance data. I presume the majority of the patients received first line therapy and there may be potential for drug resistant clusters to spread as if undetected patients may fail the standard therapy, I would ask the authors to comment on this.

6. PLOS authors have the option to publish the peer review history of their article (what does this mean?). If published, this will include your full peer review and any attached files.

Reviewer #1: No

Reviewer #2: No

---

## [Author Response · Author response to Decision Letter 0]

6 Jan 2023

Reviewer #1:

1. Response to comment: In discussion part, the author thought the prevalence of TB in Hubei has been controlled to a certain extent. However, it could be biased if it was judged only by decline of notification rate. It may be due to low detection of PTB cases. More data was needed to this conclusion. 

Response: It is really as Reviewer suggested that it could be biased to judge that the prevalence of TB in Hubei has been controlled to a certain extent only by the decline of notification rate. So I have removed the description, and described what the mainly effective measures were taken by government to control TB. 

“It showed that the prevalence of PTB in Hubei province has been controlled to a certain extent.” was deleted.

2. For season trend, the author explained that most TB infections may occur in winter, with symptoms appearing after a certain time. However, for most adults in China, TB infection more likely occurred in previous years rather than recent one year. Thus this point was not reasonable. And also for “the impact of delayed diagnosis should also be considered”, what does it mean? Need much more explanation.

Response: As Reviewer suggested that most adults TB infection more likely occurred in previous years rather than recent one year. I know that the once infected, the individual is at highest risk of developing TB disease within the first two years, and among family contacts of TB patients, children younger than 5 years of age have a 19% chance of developing TB within 2 years. But I did not find relevant information about the probability of developing TB within one year after infection. So, to be more accurate, I've removed that sentence “thus most TB infections may occur in winter, with symptoms appearing after a certain time”.

About delayed diagnosis, I revised this part. In rural and remote areas of Hubei Province, most patients chose the nearest primary health facilities for their first medical consulation rather than visited TB designated hospitals directly, but the primary health facilities usually have no capacity or poor diagnosis of TB, and may misdiagnosed patients with common diseases such as cold or fever, leading to delayed in TB diagnosis, and consequently, more PTB cases were reported in summer seasons.

3. The study showed that Enshi Prefecture and Yichang city and Tongshan County in Xianning city were the areas with high notification rate of PTB in the past 11 years. Is there any change of hotspots for these regions during the 11 years? If there was some changes, it could be very useful to understand local TB control effects.

Response: Yes, Reviewer, there is any change of hotspots for these regions during the 11 years. I gave some descriptions in the manuscript. Little changed of hotspots in Enshi Prefecture, which has the highest TB epidemic in Hubei province. Changyang Tujia Autonomous County, Zigui County and Xingshan County, which were not all hot spots from 2011 to 2014. These counties are adjacent to Enshi Prefecture. This showed that there was weakness in TB control. The change of hot spots in Xianning was described in the manuscript, and the control measures are effective.

Line 322, “which were not all hot spots from 2011 to 2014.” was added.

Special thanks to you for your good comments.

Reviewer #2:

In the limitation section line 341 – 351 I feel the authors should include the fact that as far as I am aware they do not have access to any genotyping data or drug resistance data. I presume the majority of the patients received first line therapy and there may be potential for drug resistant clusters to spread as if undetected patients may fail the standard therapy, I would ask the authors to comment on this.

Response: It is really as Reviewer suggested that there may be potential for drug resistant clusters to spread. Unfortunately, in recent years, the proportion of drug-resistant tests in Enshi and Yichang has been increasing year by year, and the number of drug-resistant patients has also been increasing, but not all patients with active TB can be tested for drug sensitivity. Since we don't have comprehensive drug susceptibility test data, we didn’t do this analysis. This should be the foucus of our future efforts. 

Line 360-362, “There may be potential for drug resistant clusters to spread. However, we did not perform this analysis because we did not have comprehensive drug susceptibility test data for all active TB patients.” was added.

Thank you very much for your good comments.

Other changes:

1. Fig 1, I changed Fig 1 to a new one, please see the attachment.

2. Line 12,169,171(table) and 174, the statement of “pathogenic positive” were revised as “bacteriologically confirmed”.

3. Line 86, the statement of “geographic database from China CDC” were revised as “We obtained vector map files from National Geomatics Center of China at a 1:1,000,000 scale as the layer’s attribute. ((https:// www.ngcc.cn/ngcc/).”

4. Line 95-100, “Ethical statement” was added, Line 153-157 was deleted. 

5. Line302 “Yichang County” was revised as “Yichang city”.

6. Line 386-391, “Supporting information” were added.

7. Line 527-534, reference No.36 was revised.

---

## [Editor Report · Decision Letter 1]

25 Jan 2023

Spatial-temporal analysis of pulmonary tuberculosis in Hubei Province, China, 2011-2021

PONE-D-22-10280R1

Dear Dr. Zhou,

We’re pleased to inform you that your manuscript has been judged scientifically suitable for publication and will be formally accepted for publication once it meets all outstanding technical requirements.

Kind regards,

Zhenlong Li, Ph.D.

Academic Editor

PLOS ONE
---

## [Editor Report · Acceptance letter]

27 Jan 2023

PONE-D-22-10280R1 

Spatial-temporal analysis of pulmonary tuberculosis in Hubei Province, China, 2011-2021 

Dear Dr. Zhou:

I'm pleased to inform you that your manuscript has been deemed suitable for publication in PLOS ONE. Congratulations! Your manuscript is now with our production department. 

Kind regards, 

on behalf of

Professor Zhenlong Li 

Academic Editor

PLOS ONE